# Exploring Sustainable Development Pathways for Agri-Food Supply Chains Empowered by Cross-Border E-Commerce Platforms: A Hybrid Grounded Theory and DEMATEL-ISM-MICMAC Approach

**DOI:** 10.3390/foods12213916

**Published:** 2023-10-26

**Authors:** Gaofeng Wang, Yanning Hou, Changhoon Shin

**Affiliations:** 1School of Management, Henan University of Technology, Zhengzhou 450001, China; wanggaofeng@haut.edu.cn; 2School of Management, Zhengzhou University, Zhengzhou 450001, China; 201917060324@stu.haut.edu.cn; 3College of Ocean Science and Engineering, Korea Maritime & Ocean University, Busan 49112, Republic of Korea

**Keywords:** cross-border e-commerce platforms, agri-food product supply chain, sustainable development, driving mechanisms, implementation pathway

## Abstract

As cross-border e-commerce platforms become increasingly integrated into the agricultural supply chain, the establishment of a sustainable supply chain ecosystem is of paramount importance. This study, grounded in the platform theory and the supply chain ecosystem theory, combines the grounded theory and the DEMATEL–ISM–MICMAC model to thoroughly analyze the complex mechanisms driving sustainable development. Utilizing the grounded theory, we construct a system of driving factors comprising five primary indicators and eighteen secondary indicators. The hybrid model reveals the interrelationships, significance, system hierarchy, and dependence-driving relationships among these factors. Notably, the driving factor system is categorized into a six-level hierarchical structure, encompassing profound elements, such as policy optimization and digital empowerment, as well as surface-level factors, such as simplification of customs procedures and consumer demand forecasting. Based on the analysis results, this research proposes a set of pathways to achieve the sustainability of the supply chain. These strategies involve improving cross-border agricultural e-commerce policy frameworks, enhancing digital-driven supply–demand coordination, strengthening logistics infrastructure and transparency, and cultivating brand influence. The study’s findings not only enrich the relevant theories but also provide practical guidance for the coordinated advancement of economic, social, environmental, and resilient development. Furthermore, they are conducive to advancing the United Nations Sustainable Development Goals.

## 1. Introduction

In today’s world, the combined forces of economic globalization and the digitalization wave are propelling the robust development of cross-border e-commerce for agricultural products. This trend not only opens up international market opportunities for agricultural products [1] but also provides essential support for achieving agricultural modernization and the global 2030 United Nations Sustainable Development Goals [2]. However, despite the immense potential in this field, it faces several significant challenges. These challenges include a lack of seamless coordination between agricultural supply chains and cross-border e-commerce supply chains, along with a shortage of skilled professionals [3]. Meanwhile, factors such as climate change [4], food security [5], and trade protectionism [6] increase the vulnerability of this supply chain, making its sustainable development an urgent industry concern [7]. Moreover, the sustainable development of cross-border e-commerce for agricultural products is crucial for all stakeholders. It not only impacts its own stability and growth [8] but also has implications for the integration of smallholder farmers into the global market, rural revitalization, and the achievement of shared prosperity [9]. Additionally, it plays a proactive role in achieving global dual carbon goals [10]. Therefore, thoroughly exploring the promoting factors and mechanisms influencing the sustainable development of cross-border e-commerce for agricultural products is of paramount importance. This in-depth analysis not only provides a theoretical and practical foundation for decision making but also fosters the coordinated development of various aspects, including the economy, society, and the environment [11]. Moreover, it offers crucial support for achieving the related sustainable development goals, such as SDG 12, SDG 9, SDG 10, and SDG 17.

However, when examining the existing research, it becomes apparent that, in terms of research content, the focus has primarily been on the overall aspects of cross-border agricultural supply chains [12,13,14,15]. However, it is important to note that cross-border agricultural supply chains can be categorized into multiple dominant types based on their characteristics and stakeholders, such as cross-border e-commerce platform-driven and logistics-park-driven models [15]. Additionally, the related influencing factors can be further categorized into various types, including driving factors, success factors, key factors, and hindering factors [16]. Clearly identifying and categorizing the driving factors of sustainable supply chain management (SSCM) is of great importance for professionals in this field, as it helps them pinpoint the primary sustainability issues, identify potential challenges, and specify the areas for improvement [16].

While some research has addressed the promoting factors of sustainable agricultural supply chains [17,18,19,20,21,22,23,24,25,26], questions regarding the promoting factors of sustainable cross-border e-commerce platform-driven agricultural supply chains remain unclear. In particular, with the increasing role of cross-border e-commerce platforms in agricultural supply chains, their impact on the sustainable development of the supply chains becomes more prominent. However, the understanding of how cross-border e-commerce platform-driven agricultural supply chains achieve sustainability—along with their inherent characteristics and operational mechanisms—remains incomplete. This gap in the knowledge, especially in the current wave of globalization and digitization, necessitates in-depth research in order to systematically analyze how cross-border e-commerce platforms specifically drive the sustainable development of agricultural supply chains, as well as the complex relationships and interactions with other influencing factors. Only through such research can we fully comprehend the essence of this emerging phenomenon and provide appropriate responses in both theory and practice to meet the new demands of sustainability in agri-food supply chains in the digital and globalized era.

Regarding the analytical methods, the existing research has primarily utilized theoretical analyses [27,28], case studies [19,23], semi-structured interviews [29], the FAHP method [14], the FAHP–FTOPSIS method [30], the ISM model [31], the SEM method [15,32], the fuzzy AHP method [18], the Grey–DEMATEL method [24], TOPSIS–AHP, AHP, COPRAS–AHP, the Borda rule, and the Copeland method [33], as well as the entropy measurement method [34]. It is worth emphasizing that not all driving factors of SSCM have the same impact on organizations. Therefore, we must pay special attention to the prioritization within the industry to help identify the urgent issues, which should be addressed first. The introduction of the DEMATEL–ISM–MICMAC hybrid method has significant advantages, as it assists in gaining a more comprehensive understanding of the importance, complex hierarchical relationships, and interactions between various influencing factors, thus providing robust support for problem solving [35].

From a theoretical standpoint, the current research has primarily been built upon foundational theories, such as the supply chain management theory, supply chain contract theory [13], biological immunity theory, organizational learning theory [15], social exchange theory, transaction cost theory [15], fuzzy theory [30], and the SCOR theory [32]. However, there has been a lack of comprehensive integration of the platform theory and the supply chain ecosystem theory as a theoretical framework in these research works. Moreover, there is a need to combine the grounded theory to further refine this framework and the corresponding indicator system. Given the unique characteristics of cross-border e-commerce platforms, introducing the platform theory and the supply chain ecosystem theory as analytical tools provides a fresh perspective for in-depth research into the sustainable development of cross-border e-commerce platform-driven agricultural supply chains. Additionally, utilizing a procedural grounded theory approach through continuous data comparison, systematic coding, and sensitive insights into the theory will be invaluable [36]. This comprehensive approach will contribute to a thorough understanding of the driving mechanisms and implementation pathways for sustainable development in cross-border e-commerce platform-led agricultural supply chains, enabling a deeper comprehension and interpretation of the complexities in this field and providing robust theoretical support for problem solving.

Based on these considerations, this study is grounded in the platform theory and the ecosystem theory, and it utilizes a procedural grounded theory approach. It combines the DEMATEL–ISM–MICMAC model to comprehensively explore the following four key questions:(1)What are the concepts and characteristics of sustainable development in cross-border e-commerce platform-driven agricultural supply chains?(2)What are the primary promoting factors for the sustainable development of cross-border e-commerce platform-driven agricultural supply chains?(3)How are the significance, hierarchical relationships, and interaction relationships of the promoting factors for sustainable development in cross-border e-commerce platform-driven agricultural supply chains constructed?(4)What practical, feasible pathways exist for achieving sustainability in cross-border e-commerce platform-driven agricultural supply chains?

The remaining content is organized as follows. Section 2 will begin by defining the concept and characteristics of the sustainable development of cross-border e-commerce platform-driven agricultural supply chains. It will also introduce the theoretical foundations used, including the platform theory and the ecosystem theory. Section 3 will elaborate on the research methods, which include the procedural grounded theory and the hybrid DEMATEL–ISM–MICMAC method. Section 4 will establish an indicator system for the driving factors, systematically integrating various driving factors to provide clear guidance for subsequent analyses. Section 5 will provide a detailed description of the research process, including data collection, processing, and analysis steps, as well as the presentation of the analysis results. Section 6 will discuss theoretical contributions and insights for practical applications. The final section will present the research findings and outlook for the future.

## 2. Conceptual Definition and Theoretical Foundation

### 2.1. Conceptual Definition and Characteristics

Building upon the existing literature on cross-border agri-food supply chains [37,38,39,40] and sustainable agri-food supply chains [41,42,43,44,45], this study aims to define and characterize the sustainability of cross-border e-commerce platform-driven agri-food supply chains from both theoretical and practical perspectives.

(1) Connotation of Sustainability in Cross-Border E-Commerce Platform-Dominated Agri-Food Supply Chains

The sustainability of cross-border e-commerce platform-dominated agri-food supply chains refers to the establishment of an end-to-end digital platform, which facilitates efficient collaboration and coordination among supply chain stakeholders. Leveraging the platform’s data and technological advantages enhances information transparency, resource utilization efficiency, risk management capabilities, and reduces unnecessary resource consumption and environmental pollution. The aim is to create a cross-border agri-food supply chain system characterized by quality assurance, environmental friendliness, responsiveness to demand, and risk mitigation, thus fulfilling consumer expectations for product quality and responsibility. This sustainability is achieved through the harmonious development of environmental protection, social development, economic benefits, and resilience dimensions. What is even more crucial is its pivotal role in achieving the United Nations 2030 Sustainable Development Goals. This encompasses fostering responsible production, driving inclusive and sustainable agri-food industrialization, reducing wealth inequality, and strengthening international cooperation and partnerships, among other objectives. 

(2) Characteristics of Sustainability in Cross-Border E-Commerce Platform-Dominated Agri-Food Supply Chains

Cross-border e-commerce platforms play a pivotal role in agri-food supply chains, including connecting supply chain entities to reduce coordination costs, providing information and big data support to mitigate decision risks, regulating supply chain operations by establishing quality and green standards, promoting sustainable production and consumption to stimulate green demand, enhancing resource utilization efficiency through comprehensive supply chain visualization, driving industry advancement through social responsibility evaluation mechanisms, and ensuring green transportation by relying on mature logistics systems. The empowerment of cross-border e-commerce platforms ensures that agri-food supply chains pursue not only economic benefits but also social responsibility and environmental friendliness, thus achieving sustainability. The specific characteristics of sustainability in cross-border e-commerce platform-driven agri-food supply chains include

Digitally driven platform: In cross-border e-commerce platform-dominated agri-food supply chains, the digital platform serves as a central engine. Advanced technologies, such as big data, blockchain, and cloud computing, are employed to gather supply chain information, facilitate seamless information flow, and promote effective collaboration among supply chain stakeholders. This digital platform reduces information asymmetry, enhances management transparency, and optimizes coordination efficiency, offering robust support for supply chain collaboration and sustainable development [46,47].Green and environmental orientation: Sustainability in cross-border e-commerce platform-dominated agri-food supply chains places a significant emphasis on green and environmental considerations. This entails efficient resource utilization, adoption of clean production methods, and promotion of circular economy practices at various supply chain stages [48]. Environmentally friendly and biodegradable packaging materials reduce waste and environmental pollution while positively impacting consumer perception. Additionally, utilizing eco-friendly transportation powered by renewable energy sources reduces carbon emissions and environmental footprint. Introducing environmental management systems and green assessments integrates green principles into supply chain management and decision-making processes, further enhancing the benefits of sustainable development [49].Quality control and safety assurance: Ensuring agri-food product quality and safety is a pivotal goal of cross-border agri-food supply chains. Establishing food safety regulations and product traceability systems enables comprehensive monitoring and traceability of agri-food products from production to transportation, ensuring compliance with quality standards and safety requirements. Furthermore, quality traceability and recall mechanisms enhance process control in key segments, thus enhancing the controllability and safety of supply chain quality [50].Demand responsiveness and innovation: Cross-border agri-food supply chains necessitate agile responses to constantly evolving market demands, requiring a high degree of responsiveness and innovation. Leveraging technologies such as big data enables accurate insights into market fluctuations, facilitating timely adjustments to production and supply strategies [51]. Establishing flexible supply chain systems to meet diverse and innovative product or service demands enhances supply chain competitiveness [52].Multi-party collaboration and win–win mechanisms: Collaboration among multiple stakeholders is paramount in cross-border agri-food supply chains. Establishing strategic partnerships, which integrate resources from various participants, leads to risk sharing and benefit sharing, ultimately enhancing the efficiency and sustainability of the supply chain. Dynamic optimization mechanisms for business and technological innovation [53], coupled with social responsibility, contribute to stable and beneficial cross-border agri-food supply chain ecosystems [54].

### 2.2. Platform Theory 

The platform theory, originally derived from the field of engineering and now widely applied in economics and business, provides a significant framework for explaining digital platforms involving multiple interactions [55]. This theory underscores the “multi-sided” nature of platforms, encompassing multiple supply, demand, and transaction facets. The four major functions of a platform include achieving participant collaboration, aggregation, interaction, and recombination [56]. In supply chain contexts, the platform theory integrates various stages into a unified digital collaboration system, facilitating efficient information and logistics flow and optimizing supply chain operations [57].

The platform theory is of particular significance in the realm of cross-border e-commerce [58]. As a digital platform connecting global suppliers, consumers, and other stakeholders, cross-border e-commerce platforms facilitate the smooth flow of goods and information [59]. This not only expands market size but also internationalizes supply chain collaboration [60]. However, the challenges related to platform governance and data privacy must be addressed. In essence, the platform theory offers a fresh perspective for studying sustainable development driven by cross-border e-commerce platforms in agri-food supply chains, aiding in identifying the driving mechanisms and implementation pathways while providing theoretical support for achieving sustainability goals.

### 2.3. Supply Chain Ecosystem Theory

The supply chain ecosystem theory views the supply chain as a complex and open ecosystem, where companies compete and interdepend [61,62]. In this system, the core enterprises act as leaders, shaping the structure and development direction of the supply chain through their decisions and actions [63]. Collaboration and resource sharing among enterprises are essential for maximizing overall benefits [64,65]. Additionally, attention to network effects and environmental adaptability is crucial for maintaining supply chain flexibility. 

Supply chain evolution follows the “variation–selection–retention” mechanism [66], emphasizing the elements of adaptive evolution. This theory employs an ecological perspective to analyze supply chain complexity [67], offering significant academic support for comprehending and optimizing supply chains.

In this study, applying the supply chain ecosystem theory offers novel perspectives, aiding in a deeper analysis of how e-commerce platforms shape supply chain structures and how different enterprises collaborate to respond to environmental changes, thus promoting sustainable supply chain development. The adoption of this theory enriches the pathways of inquiry into complex issues. In conclusion, the supply chain ecosystem theory provides a solid theoretical foundation for research, its systemic approach effectively propelling the exploration of complex mechanisms and the discovery of new paths.

## 3. Methods 

### 3.1. Programmatic Grounded Theoretical Approach 

In view of the fact that the driving mechanism and realization path of sustainable development of cross-border agri-food product supply chain led by cross-border e-commerce platforms is an emerging field, which has not been fully studied, the theoretical foundation is relatively weak, and its internal laws and mechanisms need to be further explored. The grounded theoretical approach is precisely a seminal research method dedicated to the development of theoretical frameworks in emerging research areas [68]. It constructs new explanatory theoretical models by identifying the patterns and key elements of problems in a data-driven way through open, pivotal, and selective coding of raw primary data.

In contrast, the existing studies mainly follow a quantitative empirical path, ignoring the advantages of qualitative research in finding the essence of the problem [69]. Since the grounded theoretical method does not make a priori assumptions, can continuously compare data, and form a coding system covering core categories, it is highly suitable for the exploration of complex dynamic mechanisms in this study [70]. To ensure the scientific nature of the research, we focus on the programmatic implementation of grounded theories to improve the transparency and falsifiability of the research [71]. The basic steps of the procedural rooting theory research are shown in Figure 1.

### 3.2. DEMATEL–ISM–MICMAC Method 

In this study, we will first employ the DEMATEL method to analyze the interrelationships among various factors in the sustainable development of cross-border agri-food supply chains [72]. Subsequently, the ISM method will be applied to establish a multi-level hierarchical model of these factors. Finally, the MICMAC method will be used to determine which factors have the greatest driving force and dependency [73]. Through this research method, we can scientifically identify the key elements and their mechanisms within a complex system, providing a solid methodological support for research conclusions. The specific steps are as follows:(1)Data collection: Design questionnaires and invite experts to rate the impact of each factor. Rating criteria: 0 for no impact, 1 for minor impact, 2 for moderate impact, 3 for significant impact, and 4 for very high impact [74].(2)Construct direct impact matrix Z: Process the collected questionnaire data to calculate the average and construct the direct impact matrix Z.(3)Calculate composite impact matrix T: Standardize the average direct impact matrix Z to obtain the composite impact matrix T. The unit matrix E is used for calculation.
(1)X=Zmax⁡[max1≤j≤n⁡∑i=1maij,min1≤i≤m⁡∑j=1naij]
(2)$$T=t_{ij}=X{\left(E-X\right)}^{-1},\;E\;\mathrm{represents}\;\mathrm{the}\;\mathrm{identity}\;\mathrm{matrix}.$$

(4)Analyze composite impact matrix T: Calculate the influence, influenced by centrality and causality, using the composite impact matrix T, revealing the relationships among factors [75].


(3)
$$D_{i}=\sum_{j=1}^{n}t_{ij},(i=1,2\ldots ,n)$$



(4)
$$C_{i}=\sum_{j=1}^{n}t_{ji},(i=1,2\ldots ,n)$$



(5)
$$M_{i}=D_{i}{+C}_{i},(i=1,2\ldots ,n)$$



(6)
$$R_{i}=D_{i}{-C}_{i},(i=1,2\ldots ,n)$$


(5)Calculate reachable matrix: Calculate the overall impact matrix H, derive the threshold λ by averaging all values in the composite impact matrix T, and transform the overall impact matrix H into the reachable matrix M.


(7)
$$H=T+I=h_{ij}$$



(8)
$$M={\left[m_{ij}\right]}_{n\times n},(i,j=1,2,\ldots ,n)$$



(9)
$$m_{ij}=\left\{\begin{array}{c} 0,h_{ij}<\lambda \\ 1,h_{ij}\geq \lambda \end{array}\right.$$


(6)Calculate reachable and preceding sets: Utilize the reachable matrix M to determine the reachable and preceding sets of elements. The reachable set is the collection of row vector elements, which can be reached in the reachable matrix M, and the preceding set is the collection of column vector elements, which can be reached [74].


(10)
$$R\left(s_{i}\right)\cap Q\left(s_{i}\right)=R\left(s_{i}\right)$$


(7)Construct ISM model: Based on the elements, which satisfy Formula (10), build a hierarchical structure. In each extraction, remove the used elements and repeat the process until all elements are categorized into different hierarchical levels.(8)Calculate dependency and driving force values: Sum the rows and columns of the reachable matrix M to obtain the driving force and dependency values of each factor. The driving force indicates the extent to which a factor influences other factors, while dependency indicates the degree to which a factor is influenced by other factors.


(11)
$$Q_{i}=\sum_{i=1}^{n+1}m_{ij}$$



(12)
$$Y_{i}=\sum_{j=1}^{n+1}m_{ij}$$


(9)Visualize dependency and driving force quadrant: Plot the calculated driving force and dependency values in a quadrant graph, providing a visual representation of the relationships between factors.

## 4. Index System Construction

### 4.1. Data Collection and Processing

In order to ensure comprehensiveness and scientific rigor, we gathered the relevant literature from various sources, including Chinese academic databases, such as CNKI, international databases, such as Web of Science and Elsevier, and official government websites. The literature search covered the years 2018 to 2022 and focused on keywords, such as “cross-border e-commerce supply chain impact factors”, “sustainable supply chain driving factors”, “agri-food supply chain impact factors”, and “cross-border e-commerce platform impact factors”. Additionally, policy documents from government sources and industry reports were also collected to enrich the data pool. News articles and reports from industry organizations and media outlets related to the sustainable development of cross-border agri-food supply chains and cross-border e-commerce platforms were also considered. The selected time frame of 2018 to 2022 was based on several considerations to ensure the study’s comprehensiveness and timeliness.

First, we recognized that 2018 marked a significant point with the escalation of the US–China trade war. The turbulent changes in trade policies and the international economic environment during this period directly impacted the development of cross-border e-commerce and had profound implications for the strategies and operations of e-commerce platform-driven cross-border agri-food supply chains. Second, the global pandemic outbreak in 2020 led to massive lockdowns and restrictions. During this crisis, cross-border e-commerce demonstrated remarkable resilience, experiencing rapid growth and showcasing a high degree of flexibility to adapt to new market demands. This highlighted the adaptability of e-commerce platform-driven cross-border agri-food supply chains. Third, 2022 is the most recent year for which comprehensive statistics are available, allowing us to capture the latest dynamics in the fields of cross-border e-commerce and agri-food supply chains. Selecting this time point helps us understand the current trends and changes more accurately and evaluate the sustainability factors of e-commerce platform-driven cross-border agri-food supply chains with greater precision. Moreover, focusing on a five-year data range allows us to capture the developmental trajectory of e-commerce platforms within a rapidly changing environment, considering factors such as policy changes, technological advancements, and market fluctuations. Overall, the chosen time frame from 2018 to 2022 strikes a balance between a reasonable time span and relevance to the research topic.

Based on the research theme, a strict selection process was employed to ensure the quality and relevance of the collected literature. Only literature directly related to “cross-border e-commerce”, “agri-food supply chain”, and “sustainable development” was retained, ensuring alignment with the research topic. Literature with weak content quality and low reference value was excluded to ensure the selected sources were of high quality and credibility. Additionally, for highly repetitive literature, only one version was retained to ensure diversity and comprehensiveness of the selected sources [76]. These rigorous selection criteria ensured the quality and reliability of the chosen literature.

To ensure the systematic and scientific nature of the coding analysis, a coding team consisting of interdisciplinary experts, doctoral students, and industry professionals was formed. After careful screening, a total of 358,000 words of coding research samples were collected, of which 111,000 words were not coded and were reserved for future theoretical saturation testing. Based on the remaining 247,000 words of samples, a combination of NVivo software-assisted coding and manual coding was employed. This involved progressive open, axial, and selective coding through paragraph scanning, keyword extraction, and comparisons of similarities and differences. This process aimed to identify variables and latent concepts. Eventually, 36 records related to the research topic were constructed, forming a preliminary set of indicator systems for the driving factors. This fusion of artificial intelligence and human intelligence in the coding process enhanced the preliminary set of indicator systems for the driving factors. This fusion of artificial intelligence and human intelligence in the coding process enhanced the scientific rigor of the study and ensured both in-depth exploration of the original data and the organic construction of the theoretical model [77].

### 4.2. Data Coding 

#### 4.2.1. Open Coding

Open coding involves systematically organizing and breaking down raw data, extracting concepts from them, and forming categories based on the relationships between these concepts [78,79]. In this stage, we meticulously organized the collected data, selecting conceptual statements for reassembly into conceptual categories. Due to the quantity and complexity of initial concepts and their related statements, there might be overlaps and repetitions among the initial conceptual categories. Through comparison, induction, and organization of the original data, we obtained 36 initial concepts and 18 initial categories.

#### 4.2.2. Axial Coding 

Axial coding builds upon the initial categories derived from open coding. It involves analyzing the relationships between various categories, establishing connections, and further refining, abstracting, and integrating similar categories through multiple comparisons to create higher level concepts [80]. In this study, we conducted an in-depth analysis of the refined 18 initial categories and classified them based on their logical relationships. Ultimately, we arrived at five axial codes: environmental system, comprehensive service support system, cross-border agri-food ecosystem, cross-border agri-food product logistics system, and cross-border e-commerce platform system. These axial codes facilitate a better understanding and organization of the extracted concepts, providing a clear framework for further analysis and research.

#### 4.2.3. Selective Coding

In the process of data coding, selective coding involves a deeper level of abstraction and summarization of previously refined categories to uncover the inherent logical relationships between categories. This helps in identifying a core category and associating other general categories with it, ultimately forming a concise theoretical model [79].

Through open and axial coding, we distilled five major categories: external environment system, service support system, agri-food ecological system, logistics distribution system, and e-commerce transaction system. Building upon this foundation, we recognized the e-commerce platform system as the core category, with other categories exhibiting typical functional relationships with it, and the specific categories and core category affiliations are shown in Table 1.

As the external environment, the environmental system’s changes impact various elements within the ecological system. The service support system provides foundational support, while the agri-food production system and logistics distribution system function as critical upstream and downstream components. The e-commerce transaction system acts as a nexus and driver. Specifically, the e-commerce transaction system symbiotically enhances logistics efficiency with the logistics distribution system, collaborates harmoniously with the service support system to augment supply chain coordination, propels green transformation in the agri-food production system, and guides the logistics system to reduce environmental impact. These systems collaborate and innovate synergistically around the core of the cross-border e-commerce transaction system, collectively propelling the sustainable development of the supply chain. The specific content is shown in Figure 2.

Overall, this study employs selective coding to abstract and summarize the inherent logical relationships among multiple system categories while establishing the core role of the cross-border e-commerce platform system. This enables us to construct a relational model, which delves deeply into the complex mechanisms influencing the sustainable development of cross-border agri-food supply chains. Specifically, the model reveals the interactions between different systems and their support or dependency on the core system, highlighting the systemic synergy of supply chain sustainability. This relationship exploration and system reconstruction not only deepen our understanding of intricate issues but also provide a crucial analytical tool and decision-making basis for devising strategies and policies for system optimization.

#### 4.2.4. Theoretical Saturation Test

The theoretical saturation test is a validation method used to determine whether a sufficient amount of data has been acquired to delve into the characteristics of specific categories. Its objective is to confirm the completeness and adequacy of the extracted concepts, categories, and their relationships [80]. In this study, we conducted a theoretical saturation test using the reserved 110,980 words of original data, and the results revealed no new categories or relationships. Therefore, the theoretical model constructed in this paper passed the theoretical saturation test and possesses a certain explanatory capacity. Furthermore, this study adhered to the assessment criteria of the grounded theory [75,81,82,83,84,85,86]. Rigorous quality checks were conducted along two dimensions: category saturation, theoretical completeness, operability, and theoretical quality, as well as coding reliability. These quality checks ensure the scientific rigor of the research process and the credibility of the results.

### 4.3. Construction of Driver Factor Indicator System 

Through the process of data collection and coding, we obtained five main axial codes and eighteen detailed open codes, further constructing an indicator system consisting of five primary indicators and eighteen secondary indicators, as detailed in Table 2.

## 5. Research Process

### 5.1. Data Collection and Processing 

To ensure the research objectives were achieved, this study collected expert opinions through questionnaire surveys. The criteria for expert selection were as follows: having over 4 years of experience in cross-border e-commerce, working as senior management of cross-border e-commerce companies, supply chain management researchers, and government officials, covering multiple fields, including supply chain management, e-commerce, and policy making, and being able to represent various stakeholders in the cross-border e-commerce agricultural product supply chain. Through rigorous selection, a total of 20 experts in the field were involved in the questionnaire survey, and their backgrounds are shown in Table 3. 

We employed a 0–4 scoring system to assess the degree of influence among different driving factors, with higher numbers indicating greater impact. To validate the reliability and validity of the questionnaire, we conducted Cronbach’s α test on the collected responses. The result was 0.748, surpassing the threshold of 0.7, indicating that the questionnaire exhibited good reliability and validity [84].

### 5.2. DEMATEL Model Construction

In order to further investigate the interrelationships among the driving factors for sustainable development in the e-commerce platform-led cross-border agri-food supply chain, this study employed the DEMATEL model for detailed analysis. During this process, we averaged the collected questionnaire data to generate the direct influence matrix Z. The specific content of the matrix is provided in Appendix A.

In order to conduct comprehensive impact analysis, we applied Equations (1) and (2) to standardize the data in Appendix A, resulting in the creation of the comprehensive influence matrix T, and the results are shown in Appendix A. In the subsequent analyses, the comprehensive influence matrix T plays a crucial role, as it enables us to more accurately assess the interrelationships and degrees of influence among various factors [85].

Following the calculations outlined in Equations (3)–(6), we obtained the centrality and causality values for the driving factors. Subsequently, we utilized these computed values to create graphical representations of the centrality and causality of the driving factors. These graphs visually illustrate the relationships among the various factors through a graphical format. The specific details are depicted in Figure 3.

Looking at the role of driving factors, it becomes evident that nine factors, such as policy environment optimization, financial innovation services, digital empowerment, and talent cultivation support, exhibit a degree of influence greater than 0. This suggests that their impact surpasses the influence they receive from other factors. They assume a more proactive role within the system and act as the driving forces, directly propelling the sustainability of cross-border agricultural product supply chains led by e-commerce platforms. Conversely, the remaining factors are categorized as outcome factors.

### 5.3. ISM Model Construction 

Following the steps outlined in Equations (7)–(9), we performed the calculations, established the ISM model, and generated the corresponding reachability matrix M. The results are shown in Appendix A.

Based on the analysis of the reachability matrix M, we identified the reachable sets $R\left(s_{i}\right)$ and preceding sets $Q\left(s_{i}\right)$ of element $S_{i}$. The specific results are shown in Appendix A.

Continuing, we extracted all elements, which met the criteria from the data, collectively constructing a hierarchical structure. In each subsequent round of extraction, the selected elements were removed from the candidate list to ensure avoidance of repetition. This cycle was iterated continuously until all elements were individually selected. Ultimately, a multi-layered hierarchical model was obtained, showcasing the relationships among elements at each level. The specific process is illustrated in Figure 4.

The L1 and L2 layers encompass surface factors, including S2, S3, S7, S8, and S9. These factors play a direct role in driving the sustainability of cross-border agricultural product supply chains dominated by e-commerce platforms. They have a direct impact on the chains’ stable development and are influenced both directly and indirectly by various influencing factors. The L3 and L4 layer factors are transitional elements, incorporating S13, S17, S18, S4, S6, S10, S12, S14, and S16. These factors, taken together, constitute the transitional layer in a multi-layer hierarchical structural model. They are influenced not only by deep-seated factors but also by surface-level elements, effectively serving as a bridge connecting deeper and surface-level factors. The L5 and L6 layer factors are categorized as profound elements and form the foundational core of the influencing factor system. This category primarily includes S1, S5, S11, and S15. These factors exert their influence on higher level driving factors through diverse pathways, constituting the most fundamental elements propelling the sustainability of cross-border agricultural product supply chains led by e-commerce platforms.

### 5.4. MICMAC Model Construction

Finally, calculations were performed according to Formulas (11) and (12) to determine the driving power and dependency values of each driving factor. 

Subsequently, these values were plotted on a coordinate system, where the horizontal axis represents dependency and the vertical axis represents the driving power. Each factor corresponds to a point in the coordinate system, and its specific coordinates reflect its driving power and dependency. Through the distribution of these points, we are able to classify different factors into four categories: spontaneous factors, dependent factors, linkage factors, and independent factors. Detailed results are shown in Figure 5.

The first quadrant pertains to spontaneous factors, encompassing S4, S6, S10, S14, S15, and S16. These factors exhibit relatively low levels of both dependence and driving force, being typically situated within the mid-tier of the model. They are primarily influenced by the lowest level factors and serve as essential contributors to reaching the top level. Therefore, it is imperative to give considerable attention to this quadrant. The second quadrant relates to dependent factors, which include S3, S7, S8, S9, S13, S17, and S18. The factors in this quadrant exhibit a high level of dependency and a low driving force, being often positioned at the uppermost layer of the hierarchical structural model. The third quadrant involves the linkage factor, represented by S2. Within this quadrant, the factors demonstrate a strong driving force and dependency. The fourth quadrant is reserved for independent factors, which consist of S1, S5, and S11. These factors display a high driving force and low dependency, being typically positioned at the base of the ISM model. They serve as fundamental elements within the model and exert a sustained impact on the system.

## 6. Discussion 

### 6.1. Theoretical Contribution

#### 6.1.1. Emphasizing the Stakeholder Perspective in Sustainable Agricultural Supply Chains

Based on the main entities and characteristics of cross-border agricultural product supply chains, various dominant types of supply chains can be identified [15]. However, this study focuses on the predominant role of cross-border e-commerce platforms within cross-border agricultural product supply chains. Grounded in the platform theory and the supply chain ecosystem theory, and integrating the grounded theory, we investigate the driving factors of sustainability. Firstly, we delineate the essence and features of a platform-driven sustainable supply chain for cross-border agricultural products, introducing a new perspective to the field of supply chain management. This concept not only provides a solid foundation for future research but also offers valuable theoretical support for achieving practical sustainability goals. Secondly, we extend the application domains of the platform theory and the supply chain ecosystem theory, highlighting the central role of cross-border e-commerce platforms in supply chain coordination and sustainable development. We emphasize the importance of coevolution mechanisms in achieving sustainable supply chain development. Finally, by constructing a system and model for the driving factors of a platform-driven sustainable supply chain for cross-border agricultural products, we provide new theoretical support for the expansion and deepening of theories related to factors driving sustainability in agricultural supply chains.

#### 6.1.2. Refinement and Extension of Research Scope

The existing research primarily focuses on cross-border agricultural product supply chains [2,13,14,15] and the driving factors of sustainable agricultural product supply chains [17,18,19,20,21,22,23,24,25,26]. However, there is a scarcity of studies, which organically integrate these fields. Yet, the cross-border e-commerce platforms leading cross-border agricultural product supply chains have gained prominence in today’s global trade, showing robust growth. They have become a significant channel for global agricultural trade, and achieving sustainable development in this supply chain model holds critical strategic importance.

In this study, we not only conduct a deep analysis of the key facilitating factors influencing the sustainable development of e-commerce platform-led cross-border agricultural product supply chains but also reveal their intricate interrelationships. By establishing a multi-layered model, we provide a clear representation of the connections between deep-seated and surface-level factors, offering a more comprehensive understanding of achieving sustainability. Ultimately, we construct a systematic structural model, which vividly illustrates the hierarchical relationships of these factors. This research fills gaps in the existing literature, offering new theoretical insights into the sustainable development of agricultural supply chains. More importantly, these findings provide practical guidance to relevant decision makers and practitioners, assisting them in making wiser policy and strategy decisions to promote the sustainability of global agricultural supply chains while contributing to the realization of United Nations Sustainable Development Goals.

#### 6.1.3. Innovation in Research Methods

In the field of the driving factors of sustainable agricultural product supply chains, although many studies have employed mixed MCDM methods [18,33], there is a lack of research combining the grounded theory with the DEMATEL–ISM–MICMAC model to conduct an in-depth investigation into the driving factors and their mechanisms in e-commerce platform-led cross-border agricultural product supply chains. Additionally, the existing research primarily relies on the existing literature and expert consultations when constructing indicator systems [85,86], overlooking the sensitive insights derived from continuous comparison and systematic coding of big data texts using the grounded theory.

In this study, we adopt a unique approach by seamlessly integrating qualitative and quantitative research methods. First, we employ an innovative method combining qualitative and quantitative research. Initially, using the grounded theory, we construct a driving factor indicator system to delve into the theoretical model of sustainability-promoting factors in e-commerce platform-led cross-border agricultural product supply chains. We successfully identify the key indicators, such as S1, S2, S6, S7, S10, expanding upon the existing driving factor research [16,89], providing new insights to the field. Subsequently, we utilize quantitative analysis methods, such as the DEMATEL–ISM–MICMAC model, to reveal the complex relationships and promotion mechanisms in the sustainable development of e-commerce platform-led cross-border agricultural product supply chains. The specific analysis results and discussions are as follows:

(1) Analysis of Promoting Factors in the Multi-Level Hierarchical Model

a.Surface factor analysis: These factors directly promote the sustainability of e-commerce platform-led cross-border agricultural product supply chains, directly impacting their stable development, and are subject to both direct and indirect influences from various factors. These factors mainly relate to the environmental system and cross-border agricultural ecosystem, contributing to the stability and reliability of cross-border agricultural product supply in the supply chain ecosystem, driving agricultural consumption upgrades, and enhancing the efficiency and effectiveness of the supply chain [87,88]. Therefore, taking timely and effective measures regarding these factors can directly promote the sustainability of e-commerce platform-led cross-border agricultural product supply chains.b.Transition factor analysis: These factors collectively exist as an intermediate layer in the multi-level hierarchical model of sustainability-promoting factors in e-commerce platform-led cross-border agricultural product supply chains. They are influenced not only by deep-seated factors but also by surface factors, serving as bridges connecting deep-seated and surface-level factors. These factors, while driving sustainability, also need to be comprehensively influenced by several other factors to further ensure the stability and sustainability of e-commerce platform-led cross-border agricultural product supply chains.c.Deep-seated factor analysis: These factors possess the strongest promoting power, influencing other, higher level promoting factors through different paths, constituting the fundamental elements for promoting the sustainability of e-commerce platform-led cross-border agricultural product supply chains. In particular, the role of policy environment optimization (S1) is prominent among deep-seated factors, consistent with the viewpoint mentioned in Ref [25]. Additionally, brand influence expansion (S15) also holds a significant position among deep-seated factors, aligning with the existing research, which suggests that the primary driving forces for sustainable agricultural supply chains come from media to community promotion, global market reputation enhancement [90], reputation management, and more [91]. These deep-seated factors play crucial roles in the influencing system, offering vital support for the sustainable development of e-commerce platform-led cross-border agricultural product supply chains.

(2) Analysis of the Importance of Driving Factors at Different Levels

Among the deep-seated factors at the lowest level, policy environment optimization (S1) is considered to have the highest importance, similar to the research results in Refs [88,92]. The prominence of this factor lies in the fact that optimizing a well-structured policy system can promote market standardization, enhance trade facilitation, ensure food safety, drive industrial upgrades, and foster international cooperation. Thus, this factor is attributed the highest importance among deep-seated factors, providing critical support for the sustainable development of e-commerce platform-led cross-border agricultural product supply chains [93].

Among the transition factors, talent development support (S6) is regarded as having the highest importance, aligning with the results of Ref [92]. In contrast, strengthening knowledge innovation linkage (S13) is considered relatively less important. Talent development support (S6) injects vitality into the development of the supply chain ecosystem, not only maintaining the sustainable development potential of the cross-border agricultural product supply chain but also further strengthening its stability and sustainability [94].

Among the surface factors, customs clearance process simplification (S2) is identified as the most important factor. In the supply chain ecosystem, if agricultural product imports and exports are restricted, this will affect the operation of the entire supply chain, thereby threatening the sustainability of the supply chain [95]. Meanwhile, quality standards assurance (S7) is considered less important compared to other factors at the surface level.

(3) Analysis of the Interrelationships between Promoting Factors at Different Levels

In the multi-level hierarchical model, surface factors (S3—consumption demand forecasting, S9—strategic partnership maintenance) and second-level factors (S7—quality standards assurance, S8—regional cluster development) can be considered as highly dependent and low-promotion factors. These factors have relatively poor stability and high sensitivity to other factors. Specifically, by influencing the first-level factors, they promote the “three products and one standard” construction in agricultural production and strengthen the development of modern agricultural industrial clusters, thereby promoting the sustainability of e-commerce platform-led cross-border agricultural product supply chains. There are evident interrelationships between the influencing factors of the first level, which complement each other. Simultaneously, the measures to promote the “three products and one standard” construction in agricultural production and strengthen the development of modern agricultural industrial clusters will naturally enhance the stability of agricultural partnerships, promote the cultivation of high-quality varieties, improve product quality, build a brand image, and positively impact understanding customer preferences.

The factors at the third and fourth levels primarily fall into the category of spontaneous factors with low dependence and low promoting power. However, the factors at the fifth and sixth levels—such as perfecting cross-border agricultural e-commerce policies, improving logistics infrastructure, and innovating overseas marketing methods—exhibit the characteristics of high promoting power and low dependence. These factors play a core role in driving the sustainability of e-commerce platform-led cross-border agricultural product supply chains. Additionally, these factors have close interrelationships, and their high promoting power allows them to continuously influence the top-level and second-level factors. This intricate web of interrelationships plays a crucial role in shaping the sustainability landscape of cross-border e-commerce platform-led agricultural supply chains.

In summary, this innovative method not only provides a powerful tool and approach for better understanding and addressing supply chain sustainability issues, but it also sets a new example for similar problems in other fields. This integrated approach is poised for widespread application in future research, offering new possibilities for in-depth analysis of complex issues.

#### 6.1.4. Practical Implications 

(1) Enhancing the Cross-Border Agricultural E-Commerce Policy Framework

Based on the DEMATEL–ISM–MICMAC analysis results, it is evident that policy environment optimization (S1) resides at the bottom of the multi-layer hierarchical model, exhibiting a high driving force and low dependency. It also holds the highest centrality in causation. Consequently, the recommendations based on policy environment optimization are as follows: 

Firstly, simplify the registration and entry procedures for e-commerce companies and related businesses and reduce the regulatory requirements to mitigate social inequality (SDG 10) [96]. This also helps address the challenges posed by trade protectionism (SDG 17). Secondly, reduce the compliance operating costs of e-commerce companies through tax breaks and fiscal subsidies, promoting sustainable production and consumption (SDG 12). Thirdly, establish and regulate green environmental operational standards for e-commerce companies to enhance their environmental management practices, contributing to climate goals (SDG 13) [81]. Furthermore, government agencies should provide industry guidance for cross-border e-commerce companies to help them better adapt to policy requirements, ensure the smooth implementation of policies, and enable companies to respond effectively to changes in the global economic environment. To enhance the resilience of the supply chains, policies and risk management strategies, including the establishment of risk warning mechanisms, should be improved to increase the flexibility of the supply chains in the face of global crises, such as climate change. This ensures the stability and sustainability of supply chains and mitigates the adverse effects of global crises. Lastly, actively engage in international cooperation, strengthen global partnerships, and strive to secure free trade agreements with more countries to create a favorable policy environment for e-commerce companies to expand their overseas markets (SDG 17) [97].

(2) Strengthening Digital-Driven Supply and Demand Collaboration

Based on the DEMATEL–ISM–MICMAC analysis results, digital empowerment (S5) is located at the bottom of the multi-layer hierarchical model, displaying a high driving force and low dependency. Digital empowerment also holds the highest centrality in causation. Therefore, the recommendations based on S5 are as follows:

The development potential of cross-border e-commerce platforms lies in connecting more agricultural producers and consumers, expanding the market space, and unleashing market vitality. Simultaneously, this supports various aspects of achieving the United Nations Sustainable Development Goals (SDGs). Equally important is the crucial role of digital technology in improving supply chain coordination efficiency. This promotes innovative applications of digital technology in agriculture and cross-border supply chains while emphasizing the importance of strategic partnerships to foster collaboration among supply chain enterprises, creating a healthy and sustainable supply chain ecosystem, and supporting inclusive and sustainable industrialization goals (SDG 9).

Governments play a pivotal role in achieving sustainable development, especially by encouraging cross-border e-commerce companies to build unified digital ecosystems. This includes using cutting-edge technologies, such as cloud computing, for precise demand forecasting, leveraging blockchain for end-to-end traceability of goods, and promoting innovative applications, such as electronic contracts, to reduce transaction costs. Additionally, governments should promote the establishment of digital monitoring and control systems for agricultural production and distribution within cross-border e-commerce companies. These systems ensure precise management at all stages, from cultivation and processing to warehousing and transportation [98]. These measures aim to enhance the digital collaboration among upstream and downstream supply chain enterprises, reduce information asymmetry, ensure product quality and safety, and ultimately promote inclusive and sustainable agricultural industrialization (SDG 9).

Specifically, advanced technologies, such as cloud computing, aid in achieving precise demand forecasting, reducing overproduction and resource waste. Blockchain traceability enhances transparency, promoting responsible production and consumption (SDG 12) [17,75,95]. Digital monitoring allows for end-to-end control, ensuring product quality and safety. The strengthening of these collaborative efforts will drive inclusive and sustainable agricultural industrialization (SDG 9) [99].

(3) Improving Logistics Infrastructure and Information Transparency

According to the DEMATEL–ISM–MICMAC analysis results, infrastructure improvement (S11) is located at the bottom of the multi-layer hierarchical model, demonstrating a high driving force and low dependency. It falls under the causative factors. Therefore, the following recommendations are put forward:

Governments and businesses must closely collaborate to enhance the construction of logistics infrastructure. This includes the rational planning and establishment of logistics parks, distribution centers, and other critical nodes to ensure the development of high-quality infrastructure capable of withstanding disasters. This provides a solid foundation for sustainable industrialization (SDG 9) [80,82,90,100].

Businesses should actively utilize modern technologies, such as RFID, GPS, blockchain, and others, for real-time monitoring of the goods’ movements. They can offer efficient and transparent logistics services by allowing customers to access logistics information through applications (e.g., apps), thus reducing resource waste [83]. This requires governments and businesses to jointly create interoperable logistics and supply chain information platforms as supporting systems. Additionally, the use of smart technologies to enhance warehousing and transportation efficiency helps lower logistics costs [101,102]. These measures significantly improve the level of logistics infrastructure and the transparency of logistics information, ultimately achieving responsible consumption and production (SDG 12) [3].

(4) Creating Distinctive Agricultural Product Brands and Digital Marketing Platforms

According to the DEMATEL–ISM–MICMAC analysis results, brand influence expansion (S15) is located at the bottom of the multi-layer hierarchical model, playing a crucial role in the influence system. Therefore, the following recommendations are proposed:

Governments and businesses should proactively explore the unique agricultural products from various regions and nurture agricultural product brands, which represent those specific regions [63]. When selecting the suppliers, businesses need to ensure that they have a stable product quality assurance capability [91]. Utilizing digital technology for online marketing to enhance brand recognition and reputation is essential. Formulating comprehensive online and offline brand promotion strategies helps promote the development of rural areas and regions (SDG 1). Establishing interaction and reward mechanisms with consumers is crucial to improve brand loyalty [77]. These measures maximize the impact of branding and support the entry of high-quality, distinctive agricultural products into global markets, ultimately reducing urban–rural disparities (SDG 10).

Furthermore, by nurturing regional shared brands, rural economic growth can be stimulated, and the income of farmers and smallholders can be increased, integrating them into the global marketplace. Building unique brands and expanding the sales channels helps more high-quality agricultural products access international markets, thus reducing urban–rural disparities and achieving the goal of shared prosperity. These actions drive rural development, facilitate the integration of smallholders into the global marketplace, and provide strong support for reducing poverty, achieving inclusive trade, and narrowing urban–rural disparities (SDG 1 and SDG 10).

This application-oriented approach makes the research outcomes have a profound impact, capable of advancing the sustainable development process on a global scale.

## 7. Conclusions and Future Outlook

### 7.1. Conclusions

This research focused on addressing the core issue of the sustainable development path of cross-border e-commerce platforms empowering agricultural food supply chains. This was accomplished through four key phases: conceptual definition, indicator construction, empirical analysis, and path proposal.

First, the study clearly defined the essence and characteristics of sustainability in cross-border e-commerce platform-dominated agricultural supply chains. Sustainable development in these chains refers to the establishment of end-to-end digital platforms, which enable efficient coordination between upstream and downstream actors. These platforms utilize data and technological advantages to enhance supply chain transparency, resource efficiency, risk management, and to minimize resource waste and environmental pollution. They also incorporate a sense of social responsibility, creating a supply chain system, which is characterized by quality, safety, environmental friendliness, agility, and risk control, meeting consumer expectations for agricultural product quality and responsibility. This achieves coordinated development in terms of environmental protection, social progress, economic benefits, and resilience. The key features of this sustainability include digital platform promotion, green and environmentally friendly orientation, quality control and safety assurance, demand sensitivity and innovation response, and multi-party cooperation and win–win mechanisms.

Second, based on the platform theory and the supply chain ecosystem theory, the study collected text data amounting to 358,000 words from 2018 to 2022. Utilizing a systematic grounded theory approach, it comprehensively constructed a driving factor indicator system for sustainability in cross-border e-commerce platform-dominated agricultural supply chains. This system consisted of five primary indicators and eighteen secondary indicators.

Subsequently, in conjunction with the questionnaire data from 20 experts in the field of cross-border e-commerce, the study applied the DEMATEL–ISM–MICMAC model to delve into the complex relationships between the driving factors, elucidating the influence pathways and modes. The analysis revealed that among the different hierarchical driving factors, the five most fundamental factors included simplification of customs clearance procedures (S2), digital empowerment (S5), consumer demand forecasting (S3), knowledge innovation linkage (S13), and policy environment optimization (S1). Multi-level hierarchical analysis indicated that surface-level factors encompassed the simplification of customs clearance procedures (S2), consumer demand forecasting (S3), quality standards assurance (S7), regional cluster development (S8), and maintenance of strategic partnerships (S9). The transitional factors comprised knowledge innovation linkage (S13), supply chain strategic coordination (S17), enhanced risk prevention and control (S18), financial innovation services (S4), talent development support (S6), overseas warehousing construction (S10), smooth information sharing (S12), platform governance optimization (S14), and technology application upgrades (S16). The deep-level factors included policy environment optimization (S1), digital empowerment (S5), infrastructure improvement (S11), and brand influence expansion (S15). Further analysis of the interrelationships revealed that top-level factors (S3—consumer demand forecasting, S9—maintenance of strategic partnerships) and second-level factors (S7—quality standards assurance, S8—regional cluster development) are considered highly dependent but less driving factors.

Finally, a multi-level model of promoting factors was established based on the characteristics of these factors, differentiating between deep-level and surface-level elements. The study proposed the pathways for improvement, including enhancing policies, optimizing supply–demand matching, developing infrastructure, and brand building. The research findings hold significant theoretical and practical value in promoting global sustainable development and are expected to contribute to the realization of the United Nations Sustainable Development Goals, such as responsible production, economic prosperity, alleviation of hunger, and strengthened partnerships.

### 7.2. Future Outlook

This research made a valuable contribution to advancing the sustainable development of cross-border e-commerce platform-led agri-food supply chains, but there are areas for improvement.

First, this study did not adequately examine the heterogeneity of cross-border e-commerce platforms in different countries and regions and their impact on sustainability. Future research should broaden the scope of cross-border e-commerce platform diversity investigations, delving into various types of cross-border e-commerce platforms to uncover the driving factors for sustainable agricultural supply chain development and their variations in multiple contexts. Second, there is a need to further expand the research scope to include other geographical regions, such as the Belt and Road Initiative (BRI) and the Regional Comprehensive Economic Partnership (RCEP), in order to gain a more comprehensive understanding of the current status and challenges of cross-border e-commerce platform-led agricultural supply chains in terms of sustainability. Although this study’s sample encompassed cross-border agricultural data from a global perspective, it may suffer from omissions and biases in the driving factors due to the primary source of policy texts being China. Future research can diversify the data sources to obtain a broader international perspective for a more accurate assessment. Lastly, while this study employed the integrated DEMATEL–ISM–MICMAC model, limitations in terms of the time and human resources for constructing the direct influencing factor matrix resulted in a smaller sample size. Future research should actively expand the sample data and explore more advanced data analysis methods to enhance the credibility and accuracy of the analytical results, revealing a more precise and comprehensive understanding of the factors and mechanisms promoting sustainability.

## Figures and Tables

**Figure 1 foods-12-03916-f001:**
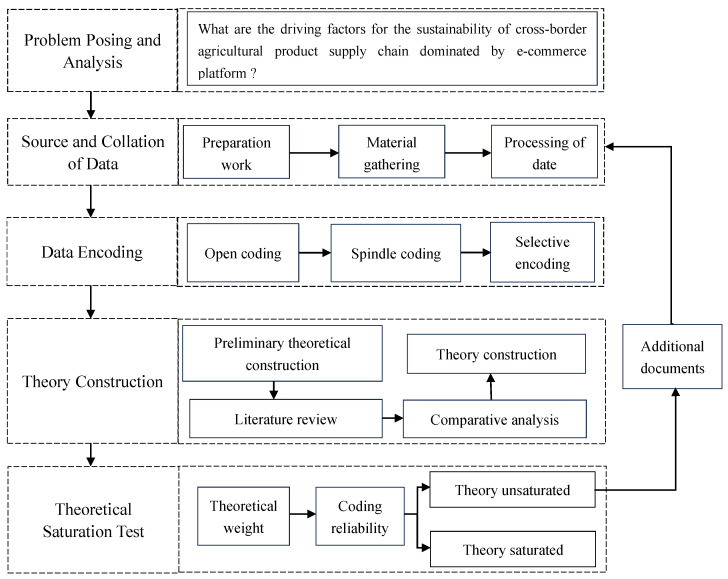
Basic steps of procedural grounded theory research.

**Figure 2 foods-12-03916-f002:**
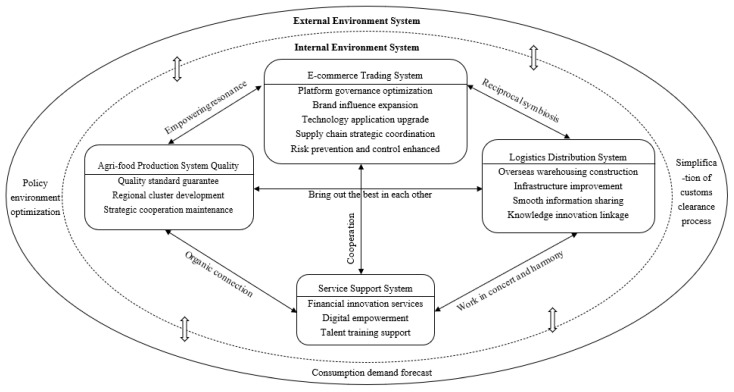
The theoretical model of sustainable drivers in e-commerce platform-led cross-border agri-food supply chains.

**Figure 3 foods-12-03916-f003:**
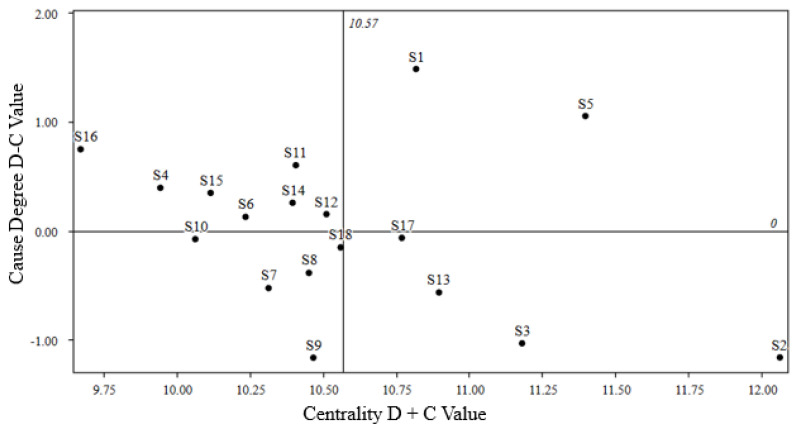
Centrality–causality diagram of driving factors.

**Figure 4 foods-12-03916-f004:**
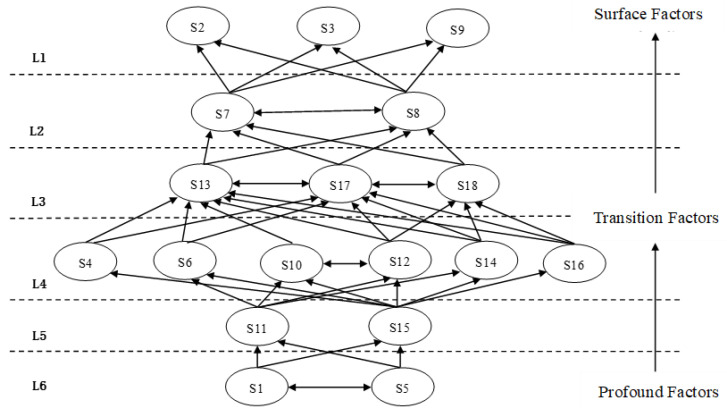
Multi-layer hierarchical structure model.

**Figure 5 foods-12-03916-f005:**
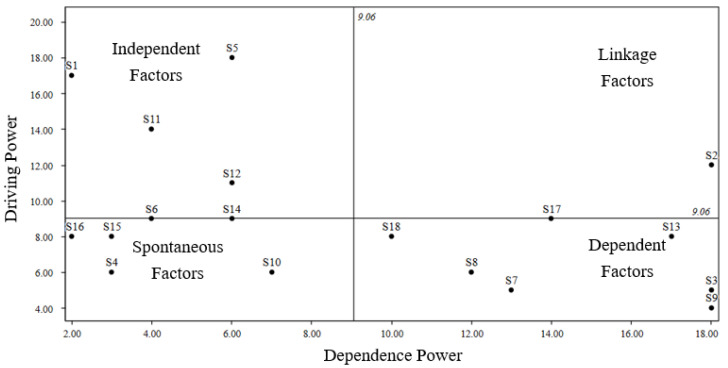
Quadrant diagram of dependency and driving power.

**Table 1 foods-12-03916-t001:** Specific categories and core category affiliations.

The Category of Spindle Code Extraction		The Initial Concept Extracted by Open Coding
Main Category	Initial Category	
External Environment System	Policy environment optimization	Improve e-commerce and agri-food product regulatory policies; Continuously optimize the regulatory environment
	Simplification of customs clearance process	Simplify customs procedures, speed up customs clearance; Reduce compliance costs
	Consumption demand forecast	Carry out consumption big data analysis; Scientifically predict consumption trends; Guide production decisions
Service Support System	Financial innovation services	Develop supply chain financial products; Provide accurate financing solutions
	Digital empowerment	Use big data, blockchain, and other technologies; Improve the efficiency of supply chain coordination
	Talent training support	Cultivate compound management and technical talents; Enhance supply chain innovation and operation ability
Agri-Food Production System	Quality standard guarantee	Implement the whole process of quality management; Ensure product quality, safety, and controllability
	Regional cluster development	Give play to the advantages of regional resources; Build an industrial cooperation ecosystem; Achieve complementary advantages
	Strategic cooperation maintenance	Establish a stable strategic cooperative relationship; Improve the stability of supply chain collaboration
Logistics Distribution System	Overseas warehousing construction	Lay out overseas warehousing network according to the market; Ensure product supply
	Infrastructure improvement	Plan logistics parks, transportation routes, and other infrastructure; Improve distribution efficiency
	Smooth information sharing	Share real-time orders, inventory, and other key information; Reduce coordination costs
	Knowledge innovation linkage	Establish knowledge sharing mechanism; Encourage technological innovation; Provide quality services
E-Commerce Trading System	Platform governance optimization	Establish a standardized operation and management mechanism; Improve platform effectiveness
	Brand influence expansion	Deepen marketing channels; Enhance brand awareness and reputation
	Technology application upgrade	Adopt safe and efficient network, data, and AI technology; Optimize the user experience
	Supply chain strategic coordination	Promote the strategic cooperation of supply chain enterprises; Achieve resource sharing and complementary advantages
	Risk prevention and control enhancement	Assess supply chain risk; Establish early warning and response mechanisms; Improve the ability to resist risks

**Table 2 foods-12-03916-t002:** Indicator system of sustainable drivers for e-commerce-led cross-border agri-food supply chain.

First-Level Indicators	Second-Level Indicators	Source
External Environment System	S1 Policy environment optimization	The construction of procedural grounded theory
	S2 Customs clearance process simplification	The construction of procedural grounded theory
	S3 Consumer demand forecast	[87,88]
Service Support System	S4 Financial innovation service	[55]
	S5 Digital empowerment	[52]
	S6 Talent training support	The construction of procedural grounded theory
Agri-Food Production System	S7 Quality standard guarantee	The construction of procedural grounded theory
	S8 Regional cluster development	[80,81]
	S9 Strategic cooperation maintenance	[75,82]
Logistics Distribution System	S10 Overseas warehouse construction	The construction of procedural grounded theory
	S11 Infrastructure improvement	[21]
	S12 Smooth information sharing	[83]
	S13 Knowledge innovation linkage	The construction of procedural grounded theory
E-Commerce Trading System	S14 Platform governance optimization	The construction of procedural grounded theory
	S15 Brand influence expansion	[1,15]
	S16 Technology application upgrade	[19]
	S17 Supply chain strategic synergy	The construction of procedural grounded theory
	S18 Risk prevention and control enhancement	The construction of procedural grounded theory

**Table 3 foods-12-03916-t003:** Sample characteristics.

Classification Project	Classification Criteria	Quantity	Proportion
Type of Personnel	Top management personnel of cross-border logistics enterprises	4	20%
	Cross-border agri-food product sustainable supply chain scientific research practitioners	9	45%
	Cross-border e-commerce platform operation and management personnel	7	35%
Years of Work	0–5 years	1	5%
	5–10 years	10	50%
	More than 10 years	9	45%
Level of Education	Junior college	1	5%
	Bachelor	2	10%
	Master of Science	10	50%
	Doctor	7	35%
Age	18–28 years old	2	10%
	29–40 years old	7	35%
	40–60 years old	10	50%
	Over 60 years old	1	5%

## Data Availability

The data used to support the findings of this study can be made available by the corresponding author upon request.

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
