# Peer review of "Exploring Sustainable Development Pathways for Agri-Food Supply Chains Empowered by Cross-Border E-Commerce Platforms: A Hybrid Grounded Theory and DEMATEL-ISM-MICMAC Approach"

_foods, 2023, doi:10.3390/foods12213916_

Round 1

Reviewer 1 Report

Authors need to address the conclusion part more strongly. How this study contributed to the literature?

How the interviewees are adequate to accomplish the research goals? It needs to ab presented at method section.

At the conclusion, authors need to elaborate how to achieve the goal of this work.

Reviewer 2 Report

The paper is in the area of sustainable development in the context of the agri-food supply chains, matching the scope of the journal.

I found the abstract to be quite long. I suggest to the authors to concentrate the information in 200 words as stated in the journal's template.

The research questions are well stated in the first section of the paper.

Section 2 and 3 of the paper are comprehensive and offer the needed knowledge for understanding the proposed approach in the paper.

The selection of the 2018-2022 period for the analysis in section 4 is presented in depth, which is more than welcomed.

Also, the information in section 5 is comprehensive, the authors are providing the information regarding the persons selected as experts for the study.

The remainder of the sections are easy to understand and follow. I think that the authors should discuss more in depth the social and economical aspects of the research paper.

Also, the authors should add few phrases related to the limitations of the study.

Minor issues:

- please refer to the sections of the paper as "sections" rather than "chapters" - please see the end of section 1.

- please use the bullets option provided by the template for the characteristics of sustainability provided in section 2.1.

- please add page numbers to the paper

- please number the equations in section 3 without indicating the number of the section.

- please try to better format table 5 - maybe using a landscape approach might help or reducing the number of digits after the comma to 2.

Reviewer 3 Report

Thank you for the opportunity to read this article. Overall is interesting and a good manuscript, after some quite major changes:

The abstract is gigantic. Must be reduced to essential about the purpose, objectives, method and findings and recommendations.

The language used it too dramatical. For example, the researcher gap is stated like this “However, its implications, driving factors, and implementation paths for sustainability remain relatively obscure”. Obscure? By the way, you should defend more adequately the need for this research explaining better what the authors 16 to 24 are bringing. You provided some (unneeded) detail about the techniques in the following lines. Do the same in explaining what the existing research is talking about (and not talking about)

Page 3 and 4 needs to be better supported on previous research. A lot of affirmations are not supported.

Tables 4 to 7 must be part of an appendix

The discussion misses the engagement with previous research. The authors must specify where their study, confirms, contradicts or advances previous research

The conclusions skyped the theoretical contributions. The authors bring two theories (Platform Theory and Supply Chain Ecosystem Theory) what is the contribution to these theories?

Round 2

Reviewer 3 Report

Congrats on the revision. Well done.